# Recent Developments of Systemic Chemotherapy for Gastric Cancer

**DOI:** 10.3390/cancers12051100

**Published:** 2020-04-28

**Authors:** Hiroyuki Arai, Takako Eguchi Nakajima

**Affiliations:** 1Department of Clinical Oncology, St. Marianna University School of Medicine, 2-16-1 Sugao, Miyamae-ku, Kawasaki, Kanagawa 216-8511, Japan; tnakajima@kuhp.kyoto-u.ac.jp; 2Kyoto University Hospital, Kyoto Innovation Center for Next Generation Clinical Trials and iPS Cell Therapy (Ki-CONNECT), 54 Kawaharacho, Shogoin, Sakyo-ku, Kyoto 606-8507, Japan

**Keywords:** gastric cancer, chemotherapy, immunotherapy, targeted therapy, biology of gastric cancer

## Abstract

Gastric cancer (GC) is a molecularly heterogeneous disease. Its molecular background, epidemiology, and standard of care are quite different between Eastern and Western countries. Many efforts have been made in developing more effective surgeries and adjuvant chemotherapies for resectable GC in each region. Recently, an intensive combination of cytotoxic agents has been established as a new standard of adjuvant treatment. Meanwhile, palliative chemotherapy is a uniform standard treatment for unresectable GC worldwide. Recently, one of the most remarkable advances in therapy for unresectable GC has been the approval of immune checkpoint inhibitors (ICIs). The use of ICIs as frontline treatment is currently being investigated. In addition, novel combinations of ICIs and targeted drugs are being evaluated in clinical trials. Despite these advances, the complex biology of GC has resulted in the failure of targeted therapies, with the exceptions of HER2-targeted trastuzumab and VEGFR2-targeted ramucirumab. GC harbors many redundant oncogenic pathways, and small subsets of tumors are driven by different specific pathways. Therefore, a combination strategy simultaneously inhibiting several pathways and/or stricter patient selection for better response to targeted drugs are needed to improve clinical outcomes in this field.

## 1. Introduction

Gastric cancer (GC) has ranked as the fifth most common cancer and the third leading cause of cancer mortality worldwide in 2018 [1]. Eastern Asia has been identified as the region having the highest incidence of GC, and Central/Eastern Europe is the region with the second highest incidence [1]. Non-cardiac GC is more frequent in these regions, whereas cardiac GC is more frequent in North America, Australia, and the UK [2]. *Helicobacter pylori* (*H. pylori*) infection is the most well-established contributing factor for GC, while Epstein-Barr virus (EBV) infection is also linked to GC development [3]. Some lifestyle factors reported to increase the risk of GC are cigarette smoking, obesity, high consumption of salt and salted preserved food, and low consumption of fruits and vegetables [3].

Despite recent advances in multimodal treatment, the prognosis remains poor for advanced GC. One of the reasons for its poor prognosis is the highly complex molecular background of GC. Many genetic and epigenetic alterations have been reported, which contribute to an aggressive phenotype of GC, such as gene mutations, differential gene expression, somatic copy number alterations, and DNA/histone methylation [4]. Given that GC is a heterogeneous disease that is likely driven by multiple genetic and epigenetic aberrations, no promising and targetable drivers have yet been identified. Although no effective treatments have been developed based on molecular characterization to date, the development of more effective treatment strategy based on new molecular data could be possible in future [5,6,7,8].

Ethnic differences are also important in considering treatment strategies for GC. Differences between cases in the Eastern countries (Asia) and the Western countries (non-Asia) include molecular features, epidemiology, and standard of care [9,10]. These differences present a barrier to the global development of superior treatments for GC and should be carefully considered. This review aims to summarize and update the recent developments in the treatment for GC. We also discuss regional differences relating to genetic background and treatment strategies, especially focusing on Asian vs. non-Asian patients.

## 2. Epidemiology

GC is difficult to intentionally detect in the early stages because it is typically asymptomatic. The high mortality of GC is primarily due to late diagnosis. Therefore, early detection and treatment is critical to reduce GC mortality [11]. Some high-risk East Asian countries have employed their own nationwide screening programs (Japan, South Korea, and Matsu Islands in Taiwan) [12]. In these countries, there is access to upper gastrointestinal endoscopy irrespective of whether an individual has symptoms. A Japanese population-based cohort study revealed that endoscopic screening can reduce GC mortality by 67% compared with radiographic screening [13]. Data from the National Cancer Screening program in South Korea showed that endoscopy was the most cost-effective screening method, which can lead to improved survival outcomes [14]. In addition, the quality of endoscopic imaging has recently and markedly improved. Image enhancement endoscopy, such as narrow-band imaging, can provide greater opportunities to detect GC earlier and allow complete endoscopic resection [15]. In fact, these active screening systems successfully led to early detection and improved survival rate. According to a nationwide population-based data in Japan between 2008 and 2009, more than 60% of GC cases were diagnosed at stage I, and 5-year relative survival was reported at 74.5% [16]. Meanwhile, Western countries do not have nationwide screening systems, resulting in later detection, unlike in Asian countries. According to the SEER-based CONCORD-2 study in the USA, the localized stage at diagnosis was reported in just 22.1% (2001–2003) or 24.9% (2004–2009) of cases, and the reported 5-year survival rate was lower than that in Asian countries (26.1% in 2001–2003 and 29.0% in 2004–2009) [17].

Tumors located in the proximal third of the stomach are more common in Western countries [18,19]. Proximal tumors are associated with more advanced stage at presentation, a larger tumor size, and poorly differentiated histology [20]. This may account for the worse survival in the West.

*H. pylori* infection increases cancer risk, especially for intestinal-type distal carcinoma [21]. The prevalence of *H. pylori* in Asia is 54.7%, which is higher than in Europe (47.0%) or in North America (37.1%) [22]. The eradication of *H. pylori* is known to result in the regression of atrophic gastritis [23]. However, the presence of intestinal metaplasia in *H. pylori*-associated chronic gastritis is suggested to be less reversible after *H. pylori* eradication than atrophic gastritis alone [24]. A meta-analysis revealed that the comparative risk of developing GC after *H. pylori* eradication was 0.65 [25]. Meanwhile, evidence showing that the cure of *H. pylori* infection reduces the risk of GC in cases of widespread intestinal metaplasia is lacking [26].

## 3. Molecular Findings in GC

GC is a molecularly heterogeneous entity, which harbors a high number of genetic alterations [27,28]. Lauren classification has originally been used to stratify GC into two types (intestinal and diffuse types) based on histological features [29]. However, it does not account for the heterogeneous nature of GC and cannot precisely predict therapeutic benefit and prognosis. Recently, The Cancer Genome Atlas (TCGA) reported a comprehensive presentation of the molecular background of GC by categorizing cases into four distinct molecular subtypes based on six different molecular platforms [5] (Figure 1). Firstly, EBV-positive tumors (9%) exhibited a higher prevalence of DNA hypermethylation, *PIK3CA* mutations, *ARID1A* mutations, and *PD-L1*/*PD-L2* amplification. A reported pathologic feature is that outstanding lymphocytic infiltration indicates activated tumor immunity in EBV-positive GC [30]. Secondly, microsatellite instability (MSI)-positive tumors (22%) showed a high mutational burden, *PIK3CA* mutations, and hypermethylation, particularly of the *MLH1* promoter. Thirdly, genomically stable (GS) tumors (20%) were enriched for Lauren’s diffuse type and showed *CDH1* mutations, *RHOA* mutations, and *CLDN18-ARHGAP* rearrangements. These genetic alterations are often associated with cell adhesion, cytoskeleton, and cell motility, resulting in an epithelial–mesenchymal transition (EMT) phenotype. Finally, chromosomal instability (CIN)-positive tumors (50%) had high somatic copy number aberrations, which were found to be associated with Lauren’s intestinal type. In CIN tumors, *TP53* mutations were common, as were amplifications of the RAS receptor tyrosine kinase pathway (*VEGFA*, *EGFR*, *ERBB2*, *ERBB3*, *FGFR2*, and *c-Met*) and cell cycle mediators (*CCNE1*, *CCND1*, and *CDK6*) [5]. Another group, the Asian Cancer Research Group (ACRG), established another classification system by stratifying GC into four subtypes based on gene expression data: microsatellite stable with TP53 functional loss (MSS/TP53-) (36%), MSS with intact TP53 (MSS/TP53+) (26%), MSS with EMT signatures (MSS/EMT) (15%), and MSI (23%) [6] (Figure 1). TCGA’s MSI group showed similarity to ACRG’s MSI subtype. Several differences were observed in other subtypes, although TCGA’s EBV-positive, GS, and CIN subtypes were somewhat enriched in ACRG’s MSS/TP53+, MSS/EMT, and MSS/TP53- subtypes, respectively [31]. This indicates that these two classifications from TCGA and ACRG are distinctive. Importantly, in contrast to TCGA, ACRG included survival data and showed the prognostic values of each subtype classification. Specifically, MSI GC has showed the best overall survival (OS) and lowest frequency of recurrence, followed by MSS/TP53+, MSS/TP53-, and MSS/EMT GC [6].

Ethnic influence on molecular characteristics has been reported [32]. Although TCGA data did not identify strong biologic differences between East Asian and other populations, some differences were observed in pathway-level gene expression changes. For example, elevated expression of the telomerase regulation pathway and decreased expression of the HIF-1-alpha transcription factor network were observed in East Asian patients [5]. Another study revealed that tumor immunity signatures significantly differed between Asian and non-Asian patients with GC [33]. Non-Asian cases of GC were associated with the enrichment of T-cell gene expression signatures and a lower expression of the immunosuppressive marker *FOXP3* compared to Asian cases of GC. To better understand the effect of ethnic differences on molecular background, further investigations with an adequate sample size are needed.

## 4. Differences in Surgical Outcomes between Eastern and Western Countries

Standard surgical procedures for resectable GC are different between Eastern and Western countries [34]. In East Asia (Japan and South Korea), radical surgery with D2 lymph node (LN) dissection has long been considered the standard. However, D1 dissection, which is less invasive than D2, is preferred in Western countries because three European randomized trials (Dutch, UK, and Italian trials) failed to demonstrate a survival benefit with D2 gastrectomy compared with D1 [35,36,37]. However, surgeons lacking experience in these studies were thought to contribute to the poor outcomes of D2 surgery. In the European randomized trials, the mortality rate after D2 gastrectomy reached over 10%, which was way much higher than that reported in the Japanese trial (0.8%) [38]. At present, the guidelines in Europe and the USA recommend D1 resection, with D2 resection being an option that should be used sparingly and only by expert surgeons in specialized and high-volume centers [39,40]. The reported frequencies of patients receiving D2 gastrectomy for resectable GC in clinical trials of adjuvant therapy were 10–55% in the West [41,42,43] and 98–100% in the East [44,45,46,47,48,49,50] (Table 1). The 5-year OS rate of patients receiving curative gastrectomy without adjuvant treatment was reported at approximately 70% in Japanese and Korean trials [51,52] and 23–35% in Western trials [36,41,42]. Of course, this discrepancy could be partly due to differences in patient characteristics among trials. However, even for the most aggressive stage (IIIB), the Asian 5-year OS rate was reported as approximately 45%, which was much higher than the overall results in the West [51,52]. This difference in surgical outcome may lead to different intensities and strategies of adjuvant therapies.

## 5. Development of Adjuvant Chemotherapy for Resectable GC

### 5.1. Western Standard Treatment

In the USA, the standard adjuvant treatment is post-operative chemoradiotherapy as per the landmark INT-0116 trial, which showed that post-operative 5-FU-based chemoradiation improved OS and recurrence-free survival (RFS) over surgery alone [41]. Most of these patients underwent resection that was less extensive than D2 (D0, 54%; D1, 36%; D2, 10%).

In Europe, peri-operative chemotherapy has been the standard, since peri-operative ECF (combination of epirubicin, cisplatin, and 5-FU) was first reported to prolong OS over surgery alone in the UK MAGIC trial [42]. Recently, Arbeitsgemeinschaft Internistische Onkologie (AIO), a German group, developed a powerful peri-operative regimen referred to as FLOT. This regimen includes three cytotoxic agents with precise dosing and scheduling (four pre-operative and four post-operative 2-week cycles of 50mg/m^2^ docetaxel, 85mg/m^2^ oxaliplatin, 200 mg/m^2^ leucovorin, and 2600 mg/m^2^ 5-FU as a 24-hour infusion on day 1). This regimen was applicable even in elderly patients [53]. They conducted a pivotal randomized phase II/III trial (FLOT4) for resectable GC, comparing peri-operative FLOT vs. ECF/ECX (epirubicin, cisplatin, and capecitabine). OS was significantly increased in the FLOT group over the ECF/ECX group (hazard ratio [HR] 0.77; 95% confidence interval [CI] 0.63–0.94; *p* = 0.012) [43]. However, in terms of safety, FLOT caused more grade 3 and 4 neutropenia (51% vs. 39%), infection (18% vs. 9%), and diarrhea (10% vs. 4%), but less grade 3 and 4 nausea (7% vs. 16%) than ECF/ECX. Based on these results, FLOT has been established as a new standard peri-operative regimen in Europe (Table 1).

Recently, more intensive strategies than peri-operative chemotherapy have been evaluated in Europe. A Dutch phase III trial (CRITICS) was conducted to evaluate a strategy to integrate post-operative chemoradiotherapy with conventional peri-operative strategy. However, the tested treatment, which consisted of pre-operative chemotherapy (ECX/epirubicin, oxaliplatin, and capecitabine (EOX)) and post-operative chemoradiotherapy, failed to improve OS compared to peri-operative chemotherapy (ECX/EOX) [54]. In this study, only 60% of all the randomized patients could receive the planned post-operative treatment, which was consistent with the MAGIC trial. This limited opportunity for post-operative treatment and the expectation of tumor downstaging and resectability after pre-operative treatment led to the successor randomized phase II trial (CRITICS-II: NCT02931890), which evaluates pre-operative chemotherapy and chemoradiotherapy. Additionally, a strategy integrating pre-operative chemoradiotherapy with conventional peri-operative chemotherapy (ECF/ECX/EOX or FLOT) is currently being evaluated in an international and intergroup phase III trial (TOPGEAR: NCT01924819) conducted in Australia, Canada, and Europe (Table 2).

### 5.2. The Standard Treatment in the East Asia

After a positive result in the Japanese ACTS-GC trial in 2007, post-operative oral S-1 for 1 year has been a standard treatment for curatively resected stage II/III GC in Japan [44]. However, the efficacy of S-1 was unsatisfactory in patients with stage III GC (5-year OS rate: 67.1% in stage IIIA and 50.2% in stage IIIA). Recently, Japan Clinical Cancer Research Organization (JACCRO) conducted a phase III trial (START-2, JACCRO GC-07) to test the superiority of S-1 plus docetaxel (DS) compared to S-1 alone for stage III GC patients. In the DS group, one 3-week cycle of S-1 was administered on day 1–14, followed by six cycles of S-1 on day 1–14 and 40 mg/m^2^ of docetaxel on day 1, given triweekly. This was followed by S-1 on day 1–28 every 6 weeks, which was continued for up to 1 year. Based on the second interim analysis, which demonstrated the superiority of DS in the primary endpoint of RFS (HR 0.632; 99.99% CI 0.400–0.998; *p* < 0.001), this trial was terminated in 2017 [45]. All adverse events in the DS arm were manageable and well tolerated. Consequently, post-operative DS has become a new standard for the curatively resected stage III GC in Japan.

In South Korea, China, and Taiwan, another pivotal phase III trial (CLASSIC) was conducted to test post-operative CAPOX (eight cycles of 1000 mg/m^2^ oral capecitabine twice daily on day 1–14 and 130mg/m^2^ of intravenous oxaliplatin on day 1, given triweekly) compared to surgery alone for stage II/III GC patients. The primary endpoint of disease-free survival (DFS) was met, and long-term follow-up data reported in 2014 confirmed the superiority of CAPOX in improving OS [46,52]. Therefore, CAPOX has been established as another standard platform for post-operative chemotherapy for stage II/III GC in Asia.

Recently, a Korean phase III trial (ARTIST-II) tested two post-operative strategies (S-1 plus oxaliplatin (SOX) for 6 months and chemoradiation added to SOX (SOX+RT)) compared to post-operative S-1 in LN-positive, stage II/III GC after curative D2 resection. As first reported at the American Society of Clinical Oncology annual meeting in 2019, post-operative SOX significantly prolonged DFS over S-1 with well tolerated toxicities at interim analysis. This was sufficient evidence for the independent data monitoring committee to stop the trial [48]. Despite post-operative chemoradiotherapy’s promising benefit for patients with LN-positive GC in the preceding ARTIST trial, no other benefit was observed in the SOX+RT arm in the ARTIST-II trial [48].

Alternatively, positive results have been reported in recent Asian phase III trials on peri-operative chemotherapy for patients with locally advanced, D2-resected GC. In the Korean PRODIGY trial, peri-operative chemotherapy consisting of pre-operative DOS (docetaxel, oxaliplatin, and S-1) and post-operative S-1 showed longer progression-free survival (PFS, primary endpoint) over post-operative S-1 in cT2/3-LN-positive or cT4 resectable GC [49]. In the Chinese RESOLVE trial, peri-operative SOX significantly improved DFS compared to post-operative CAPOX in cT4a-LN-positive or cT4b-resectable GC [50] (Table 1).

From these findings, post-operative chemotherapy has been considered the standard for curatively D2-resected GC in East Asia. In addition to two traditional regimens (S-1 for 1 year and CAPOX for 6 months), two new standard treatments (DS and SOX for 6 months) have emerged for stage III and/or LN-positive GC. Post-operative chemoradiotherapy is not a validated strategy after D2 gastrectomy. Despite the proven efficacy, it is necessary to confirm OS results after a long-term follow-up period in the PRODIGY and RESOLVE trials before introducing peri-operative chemotherapy into clinical practice for D2-resected GC patients. A Japanese phase III trial (NAGISA: UMIN000024065) testing peri-operative chemotherapy (pre-operative SOX and post-operative S-1/DS vs. post-operative S-1/DS) is ongoing, which may provide further evidence for considering this strategy.

### 5.3. Targeted Therapy for Resectable GC

Two targeted therapies currently used in clinical practice for advanced GC are being tested for resectable GC: angiogenesis-targeted and HER2-targeted therapies (Table 1 and Table 2). The MRC ST03 trial is a British phase II/III trial, which evaluates the angiogenesis-targeted therapy in resectable GC. In this trial, no survival benefit was observed with the addition of the anti-VEGF antibody bevacizumab to peri-operative ECF [55]. Currently, a German phase II/III trial (RAMSES: NCT02661971) is ongoing to test the efficacy of adding the anti-VEGFR2 antibody ramucirumab to standard peri-operative FLOT.

Given a validated survival benefit in HER2-positive advanced GC patients [56], HER2-targeted treatment is currently being tested in HER2-positive resectable GC. In Europe, two randomized trials are ongoing: INNOVATION (NCT02205047), a phase II trial evaluating the efficacy of adding trastuzumab (anti-HER2 antibody) with/without pertuzuzumab (anti-HER2 antibody) to peri-operative XP/FP (5-FU and cisplatin); and PETRARCA (NCT02581462), a phase II/III trial evaluating the efficacy of adding trastuzumab with pertuzumab to peri-operative FLOT. In Asia, a Japanese randomized phase II trial (Trigger: UMIN000016920) is ongoing to evaluate pre-operative use of trastuzumab as a strategy of peri-operative therapy for HER2-positive, extensively LN-positive, resectable GC (Table 2).

### 5.4. Immunotherapy for Resectable GC

Recently, immune checkpoint inhibitors (ICIs), which enhance antitumor T-cell activity through the inhibition of immune checkpoints such as PD-1 and CTLA-4, have successfully achieved great benefits in the treatment of various solid malignant diseases [57]. Most of these achievements have been realized in patients with unresected, advanced stage cancer. However, the adjuvant use of ICIs, such as nivolumab (anti-PD-1 antibody) and ipilimumab (anti-CTLA-4 antibody), has shown significant benefit in reducing recurrence in resected melanoma [58,59]. Recently, clinical trials have been launched to investigate the role of ICIs in the adjuvant treatment for resected GC. Currently, there are two ongoing phase III trials. First, the KEYNOTE-585 trial (NCT03221426) is evaluating peri-operative use of pembrolizumab (anti-PD-1 antibody) with chemotherapy. Surprisingly, this trial is being conducted under an international collaboration, which was thought to be difficult because there are no globally accepted standard adjuvant treatments. Considering the preference in each region, both peri-operative doublet (XP or FP) and peri-operative triplet (FLOT) regimens are allowed as backbone chemotherapy regimens in this trial. Second, the Asian ATTRACTION-05 trial (NCT03006705) is comparing S-1/CAPOX plus nivolumab vs. S-1/CAPOX plus placebo as post-operative treatments for stage III, curatively resected GC. Furthermore, two randomized phase II trials are currently ongoing in Germany: the DANTE trial (NCT 03421288) evaluating peri-operative use of atezolizumab (anti-PD-L1 antibody) combined with FLOT and the IMAGINE trial (NCT04062656) evaluating three immunotherapy arms compared to FLOT as peri-operative treatments (nivolumab, nivolumab plus ipilimumab, and nivolumab plus relatlimab (anti-LAG3 antibody)) (Table 2).

## 6. Development of Chemotherapy for Unresectable GC

The standard chemotherapy for advanced GC has been almost unified worldwide: a combination of fluoropyrimidine and platinum compounds, with trastuzumab added for HER2-positive GC, in a first-line (1L) setting and paclitaxel with or without ramucirumab in a second-line (2L) setting [39,40,60]. Monotherapies with taxane, irinotecan, and ramucirumab are also standard options of 2L treatment, which confer significant survival benefit [61,62,63,64]. Recently, trifluridine/tipiracil (FTD/TPI) and ICIs have emerged as novel standard treatments that confer validated clinical benefits in the third-line (3L) or later settings [65,66,67]. Of note, the rate of receiving post-progression therapy is much higher in Japan than in the Western countries (2L: 69–85% vs. 11–59%; 3L: 58–90% vs. 8–29%) [68,69,70,71,72].

### 6.1. Cytotoxic Agents for Advanced GC

#### 6.1.1. Triplet Regimen in 1L Treatment

Adding a third cytotoxic agent is not clinically established as standard for advanced GC. Epirubicin is conventionally used in Europe; however, its advantage is controversial [73]. Conversely, the use of docetaxel has been evaluated in clinical trials, and several docetaxel-based triplet regimens have been developed.

The first phase III trial testing docetaxel-based triplet regimens was the V325 trial conducted in the USA. In this trial, the triplet DCF regimen (75 mg/m^2^ of docetaxel, 75 mg/m^2^ of cisplatin, and 5-FU 750mg/m^2^/day (5 days), every 3 weeks) demonstrated a significant improvement in OS as a 1L treatment compared to a doublet CF regimen (cisplatin and 5-FU) [74]. However, the DCF regimen was linked to increased severe toxicities including myelosuppression and infectious complications (febrile neutropenia in 29%). Therefore, this regimen has not been used extensively in clinical practice.

There are challenges to clinically adopting a triplet regimen. The first is dose modification of the original DCF regimen. Some dose-modified DCF (mDCF) regimens were more tolerable than the original DCF and thus have become acceptable 1L options for patients with advanced GC [75,76]. Another promising attempt at controlling the severe toxicities of a triplet regimen is to replace cisplatin with oxaliplatin. FLOT and TEF, having a slightly different 5-FU dose compared to FLOT, are oxaliplatin-containing triplet regimens showing favorable toxicities [77,78]. Currently, TFOX, which is the same regimen as TEF, is being investigated compared to FOLFOX in 1L treatment in a French phase III trial (GASTFOX: NCT03006432).

In Japan, a phase III trial (JCOG1013) was conducted to test another triplet regimen consisting of docetaxel, cisplatin, and S-1 (DCS). However, DCS failed to improve OS in patients with chemotherapy-naïve advanced GC compared to S-1 plus cisplatin, which is a standard 1L treatment in Japan [79]. The dose of docetaxel in the DCS regimen was 40mg/m^2^ every 4 weeks, which was lower than that of the other effective triplet regimens. This is a possible reason for the lack of an additional effect of docetaxel in the JCOG1013 trial. In addition, overlapped renal toxicity between oral fluoropyrimidine and cisplatin is a conceivable risk in triplet regimens. Adding docetaxel increases toxicity and may change the pharmacokinetics of S-1 and cisplatin, which vary depending on the renal function of an individual. Therefore, dose adjustment may have been more difficult when DCS regimen is in use. Given these difficulties, the usage of non-oral fluoropyrimidine and oxaliplatin may be favorable in a triplet regimen.

#### 6.1.2. Trifluridine/Tipiracil (FTD/TPI, TAS-102)

FTD/TPI is a novel oral fluoropyrimidine. Its mechanism of action is unique. Specifically, trifluridine is incorporated into DNA, which results in DNA dysfunction. Tipiracil blocks trifluridine degradation by thymidine phosphorylase, increasing trifluridine’s antitumor activity. This agent has been approved for the treatment of patients with metastatic colorectal cancer that is refractory to standard chemotherapies [80]. Notably, this agent was effective even for patients that were refractory to 5-FU. Recently, in an international phase III trial (TAGS), FTD/TPI monotherapy significantly improved OS (HR 0.69; 95% CI 0.56–0.85; two-sided *p* = 0.00058) compared to placebo in advanced GC that was refractory to at least two previous chemotherapies [65]. Consequently, FTD/TPI was approved for advanced GC as a 3L or later treatment by the Food and Drug Administration (FDA) in the USA. FTD/TPI is currently under further investigation in combination with ramucirumab (NCT03686488) and irinotecan (NCT04074343 and UMIN000031346) in phase I and II trials.

### 6.2. Targeted Agents for Advanced GC

GC frequently harbors genetic alterations [4]. For example, approximately 37% of GCs exhibit amplifications of genes encoding targetable receptor tyrosine kinases (RTKs) such as EGFR, FGFR2, HER2, and MET [81]. Another potentially targetable pathway is that involving angiogenesis, which is typically altered in malignant diseases and considered to be a hallmark of GC development [4,27]. In addition, molecular pathways relating to DNA damage repair, cancer stemness, and tumor microenvironment are potential therapeutic targets in GC [4,27]. Although many targeted therapies have been tested in clinical trials, only those against HER2 and VEGFR2 have been shown to confer a modest survival advantage in advanced GC [56,64,82] (Table 3).

#### 6.2.1. HER2-Targeted Therapies

The incidence of HER2 positivity in GC according to immunohistochemistry (IHC) and fluorescence in situ hybridization (FISH) is around 20% [56]. The ToGA trial showed a significant OS benefit of trastuzumab combined with chemotherapy (XP or FP) in untreated HER2-positive advanced GC [56]. Consequently, the 1L use of trastuzumab has become the standard in HER2-positive GC.

However, other HER2-targeted therapies failed to show significant survival prolongation. For example, lapatinib, which is a small molecule tyrosine kinase inhibitor (TKI) of HER1 (EGFR) and HER2, lacked significant antitumor efficacy in HER2-positive advanced GC in both 1L (TRIO-013/LOGiC trial) and 2L (TyTAN trial) settings [83,84]. Pertuzumab is a HER2-targeted antibody that binds to a different epitope on the HER2 receptor protein than trastuzumab. Since adding pertuzumab to trastuzumab and chemotherapy improves survival in HER2-positive breast cancer [85], the JACOB trial was recently conducted to evaluate the same strategy in untreated HER2-positive GC. However, adding pertuzumab to trastuzumab and chemotherapy failed to show significant survival improvement [86]. This failure may have been related to the underlying biology of HER2-positive GC as previous studies have shown heterogeneous HER2 IHC staining patterns and lower HER2 expression in GC compared to those in breast cancer [87,88]. Even with strong HER2 positivity, inter-tumor heterogeneity suggests that some tumors within the same patient may be driven by pathways other than HER2 signaling. Thus, current HER2 testing may not completely enrich selection for tumors that are driven predominantly by the HER2 pathway.

Antibody-drug conjugates (ADCs) represent a promising class of drugs with a wider therapeutic window than conventional chemotherapeutic agents due to their efficient drug delivery to antigen-expressing tumor cells. Trastuzumab emtansine (T-DM1) is an ADC comprised of trastuzumab linked to the tubulin inhibitor emtansine by a stable linker. T-DM1 has been reported to provide significant survival benefit in patients with HER2-positive metastatic breast cancer who progressed after HER2-targeted therapy [89,90]. However, in the GATSBY trial, 2L T-DM1 failed to prolong OS compared to taxane in previously treated HER2-positive GC [91]. This failure may have resulted from the greater heterogeneous biology in GC compared to breast cancer. T-DM1 does not have a robust bystander effect due to its non-cleavable linker, and therefore may not strongly affect adjacent tumor cells that do not express the target. Trastuzumab deruxtecan (DS-8201a) is a novel HER2-targeted ADC comprised of a humanized monoclonal anti-HER2 antibody, a cleavable peptide-based linker, and a topoisomerase I inhibitor payload. The engineering of a high drug-to-antibody ratio and cytotoxic bystander effects induced potent preclinical antitumor activity across a wide range of tumor cells with various degrees of HER2 expression [92,93]. Recently, a phase I dose-escalation and dose-expansion trial showed promising efficacy of DS-8201a in heavily treated, HER2-positive GC. Surprisingly, the confirmed objective response and disease control rate were 43.2% and 79.5%, respectively. These results were observed despite all enrolled patients (n = 44) having been previously treated with trastuzumab and 44% having been previously treated with irinotecan [94]. Currently, a randomized phase II trial is being conducted in Japan and South Korea (DESTINY-Gastric01: NCT03329690) to assess the efficacy and safety of DS-8201a compared to physician’s choice chemotherapy in salvage-line treatment for HER2-positive and trastuzumab-treated GC. This trial also includes two nonrandomized cohorts for the evaluation of the efficacy and safety of DS-82001a in HER2-low and trastuzumab-naïve GC.

#### 6.2.2. VEGF/VEGFR-Targeted Therapies

In contrast to the disappointing outcomes in trials using bevacizumab as a 1L treatment for advanced GC [95,96], ramucirumab as a 2L treatment has been shown to successfully improve OS, both in combination with paclitaxel and as a monotherapy [64,82]. Based on positive results in the RAINBOW trial, paclitaxel plus ramucirumab is globally considered to be a standard 2L treatment [82]. Ramucirumab monotherapy is a 2L treatment option, given its survival benefit over placebo as per the REGARD trial [64]. However, 1L use of ramucirumab failed to improve OS in a recently reported phase III trial (RAINFALL) comparing ramucirumab to placebo in combination with XP/FP [97]. Additionally, adding ramucirumab to 1L SOX did not prolong PFS or OS in an Asian phase II trial (RAINSTORM) [98].

Apatinib is a small molecule TKI that selectively inhibits VEGFR2. In a placebo-controlled Chinese phase III trial, apatinib significantly improved OS in 3L or later treatment [99]. However, in an international phase III trial (ANGEL) conducted outside of China, apatinib failed to improve OS in the same clinical setting [100]. Therefore, apatinib may only be considered for 3L or later treatment in China.

Overall, the only anti-angiogenesis agent demonstrating significant efficacy in phase III trials for advanced GC was ramucirumab. The efficacy of anti-angiogenesis agents in GC differs from that in advanced colorectal cancer (CRC), in which continuous use of bevacizumab in combination with chemotherapy from 1 to 2L is a standard treatment strategy [101]. This disparity may be due to differences in biology between GC and CRC, including the tumor microenvironments and molecular pathways of resistance to VEGF inhibition (e.g., FGF, PIGF, and PDGF signaling) [102]. Further investigations are needed to determine the molecular background underlying angiogenesis processes in GC and identify biomarkers enabling patient selection for anti-angiogenesis treatment.

#### 6.2.3. Other Targeted Therapies

Targeting RTKs other than HER2 and VEGFR2 with monoclonal antibodies or small molecule inhibitors has resulted in disappointing efficacy so far. Anti-EGFR antibodies (cetuximab and panitumumab) have not demonstrated survival benefit in unselected patients, which was consistently demonstrated in two large phase III trials (EXPAND and REAL3) [103,104]. The reported biological features of GC suggest the existence of a small subset that is mainly driven by the oncogenic EGFR pathway [81,105]. Thus, without patient selection, this population was not enriched, which may have led to the negative results in the EXPAND and REAL3 trials.

In contrast, HGF/MET-targeted treatments were evaluated in patient-selected phase III clinical trials (RILOMET-1 and METGastric) [106,107]. However, HGF/MET-targeted antibodies (rilotumumab and onartuzumab) failed to improve survival in GC that was IHC-positive for MET expression. The MET axis is most frequently activated genomically through *MET* gene amplification, which leads to a greater MET overexpression than mere overexpression in the absence of *MET* amplification [108]. Thus, MET IHC assay may not appropriately select for MET-driven GC. This may have been one factor in the failure of these trials.

Several agents targeting pathways other than RTKs have been tested in phase III trials. Among these, mTOR, PARP, STAT3, and MMP9 inhibitors did not meet their primary endpoints [109,110,111,112,113]. None of these trials were designed to select patients based on biomarkers (Table 3).

Recently, promising results were reported in a randomized phase II trial (FAST) showing tight junction protein claudin 18.2 (CLDN18.2)-targeted zolbetuximab improved PFS (HR 0.47; 95% CI 0.31–0.70; *p* = 0.0001) and OS (HR 0.51; 95% CI 0.36–0.73; *p* = 0.0001) in combination with 1L EOX, in patients with CLDN18.2-positive GC [114]. Currently, two phase III trials are ongoing to test the 1L use of zolbetuximab in combination with mFOLFOX6 (SPOTLIGHT: NCT03504397) or CAPOX (GLOW: NCT03653507).

Overall, the development of targeted treatments for GC has failed, except for that of HER2- and VEGFR2-targeted drugs. This is likely because of the molecular complexity and heterogeneity of GC. Heterogeneity is a significant obstacle in developing treatment. One solution may be the use of circulating tumor DNA (ctDNA) to detect genetic alteration for patient selection, which may capture the tumor heterogeneity that is not detected using tumor sampling and analysis [115]. Currently, a phase III trial (FIGHT: NCT03694522) is ongoing to evaluate 1L use of the FGFR2b inhibitor bemarituzumab in FGFR2b-overexpressing GC, as determined by IHC assay, or in *FGFR2*-amplified GC, as detected using ctDNA analysis.

### 6.3. ICIs for Advanced GC

GC tumors are typically “immunologically hot” and harbor the fifth highest amount of somatic mutations among all major cancer typesm as per a large study utilizing whole-exome sequencing [116]. High mutational burden is associated with susceptibility to recognition by the immune system. GC cells develop an immune evasion system by upregulating immune checkpoint proteins such as PD-L1. PD-L1 is overexpressed in up to 65% of GC [117,118]. The PD-L1 receptor PD-1 is also upregulated on T-cells in GC patients [119]. These findings have provided a rationale for immunotherapy with anti-PD-1/PD-L1 drugs in many recent clinical trials in advanced GC. In the clinical setting, PD-L1 expression as measured by IHC assay has been assessed to define PD-L1-positive GC. Two measures of PD-L1 positivity are currently used: the tumor proportion score (TPS), which is the percentage of tumor cells with membranous PD-L1 expression, and combined positive score (CPS), which is the percentage of PD-L1 positive cells (tumor cells, macrophages, and lymphocytes) among all tumor cells. In reported GC clinical trials, tumors with TPS ≥ 1% (20–30% of advanced GC) or CPS ≥ 1% (50–60% of advanced GC) were defined as PD-L1-positive. These measures have been proposed as potential biomarkers to predict the efficacy of ICIs [67,120,121,122,123,124].

#### 6.3.1. PD-1/PD-L1 Inhibitors as Monotherapies or Combination Therapies with Cytotoxic Agents

A growing body of clinical evidence has revealed that monotherapy with PD-1/PD-L1 inhibitors has a role to play in advanced GC. It was first shown that monotherapy with PD-1 inhibitors (pembrolizumab and nivolumab) exhibited significant clinical efficacy in 3L or later treatment [66,67]. Of note, these drugs can induce marked and durable responses in around 10% of patients, while the remaining patients (~90%) did not respond. Despite the durable clinical benefit, using ICIs is associated with a spectrum of adverse events (immune-related adverse events: irAEs) that are quite different from those of the other systemic chemotherapies. IrAEs can affect multiple organs including the skin, gastrointestinal tract, lung, endocrine, thyroid, renal, and cardiovascular systems. Most irAEs are of low frequency (<5%) and manageable, but they require continuous and careful monitoring [125]. In a cohort of the global large phase II trial (KEYNOTE-059), pembrolizumab showed a higher objective response rate (ORR, 15.5% vs. 6.4%) and longer duration of response (DOR, 16.3 vs. 6.9 months) in PD-L1-positive (CPS ≥ 1%) GC than those in PD-L1-negative GC in 3L or later treatment. As a result, the FDA granted immediate approval of using pembrolizumab for 3L or later treatment of PD-L1-positive advanced GC in the USA [67]. In the Asian phase III trial (ATTRACTION-2), nivolumab significantly increased OS compared to placebo in 3L or later treatment of patients with advanced GC, leading to approved use of nivolumab in Japan, South Korea, and Taiwan [66]. Of note, the benefit from nivolumab was observed regardless of PD-L1 expression (TPS ≥ 1% or < 1%) in the ATTRACTION-2 trial, supporting its use for non-selected patients and the need for further investigation of adequate predictive biomarkers.

In contrast to the positive results in the placebo-controlled ATTRACTION-2 trial, negative results were found in two phase III trials, which evaluated the efficacy of monotherapy with PD-1/PD-L1 inhibitors compared to chemotherapy. In the JAVELIN Gastric 300 trial, 3L avelumab (PD-L1 inhibitor) failed to improve OS over physician’s choice of chemotherapy [120]. Additionally, 2L pembrolizumab monotherapy also failed to improve OS of patients with PD-L1-positive (CPS ≥ 1%) GC compared to paclitaxel in the KEYNOTE-061 trial of mostly non-Asian patients [126]. In this trial, a unique Kaplan-Meier OS curve was observed, showing that the paclitaxel group outperformed the pembrolizumab group for the first 8 months but thereafter inverted with sustained separation. This phenomenon suggested that a favorable and durable benefit from pembrolizumab may be limited to a small subset of patients, while paclitaxel was more effective for the remainder of patients. In fact, subgroup analyses showed that more pronounced benefit from pembrolizumab over paclitaxel was observed in performance status 0, CPS ≥ 10%, and MSI-high subsets. Further exploration for predictive biomarkers is a must to determine which patients should be given pembrolizumab instead of conventional chemotherapy. Currently, a similar phase III trial is ongoing in an Asian population (KEYNOTE-063: NCT03019588) to compare 2L pembrolizumab vs. paclitaxel.

Preclinical findings showing increased immunogenicity after using cytotoxic agents provide a rationale to challenge ICI as a 1L combination treatment [127]. Following the earlier phase II trial (KEYNOTE-059, cohort 2), which showed antitumor activity and tolerability of 1L treatment with pembrolizumab plus FP/XP [124], a phase III KEYNOTE-062 trial was performed. In this trial, patients with untreated HER2-negative and PD-L1-positive (CPS ≥ 1%) advanced GC were randomly allocated to three arms with two comparisons: pembrolizumab monotherapy (non-inferiority) vs. FP/XP plus placebo and FP/XP plus pembrolizumab (superiority) vs. FP/XP plus placebo. Non-inferiority of pembrolizumab was demonstrated, but superiority of FP/XP plus pembrolizumab was not demonstrated in terms of OS [128]. In the subgroup analyses of this trial, only MSI-H patients (7% of enrolled patients) showed a favorable benefit from the addition of pembrolizumab over placebo [129]. Adding pembrolizumab was tolerable with known toxicities being slightly increased. Currently, several other phase III (or II/III) trials are ongoing to evaluate 1L use of PD-1/PD-L1 inhibitors in combination with standard doublet chemotherapy (Table 4).

Based on the earlier evidence of durable activity of 1L maintenance by avelumab [121], a phase III trial (JAVELIN Gastric 100: NCT02625610) was conducted to evaluate avelumab switch maintenance treatment compared to continuation of 1L CAPOX/FOLFOX. In the latest international conference, negative results were reported, suggesting that ICI switch maintenance in 1L treatment should not be recommended [130].

#### 6.3.2. PD-1/PD-L1 Inhibitor Combination Therapy with Other ICIs

Combining PD-1/PD-L1 inhibitors with other ICIs is a promising strategy, as several immune checkpoint pathways regulate the antitumor immune response. For example, CTLA-4 is a distinct immune checkpoint molecule suppressing T-cell proliferation early in the immune response, whereas PD-1 suppresses T-cells later in the immune response [131]. In the phase I/II CheckMate-032 trial evaluating the therapeutic approach of PD-1 inhibitor combined with CTLA-4 inhibitor, nivolumab (1 mg/kg) plus ipilimumab (3 mg/kg) manifested encouraging antitumor activity and a manageable safety profile in patients with advanced GC [123]. Currently, in an ongoing phase III trial (CheckMate-649: NCT02872116), this combined regimen is being tested as an experimental arm in the 1L treatment for HER2-negative advanced GC. LAG-3 and IDO1 are other immune checkpoints currently in several phase II advanced GC trials for concurrent inhibition with that of PD-1/PD-L1 (Table 4).

#### 6.3.3. PD-1/PD-L1 Inhibitor Combination Therapy with Targeted Drugs

Recently, investigation using PD-1/PD-L1 inhibitors and targeted drugs has begun in advanced GC (Table 4).

Trastuzumab has immune mechanisms of action involving innate and adaptive immunity through antibody-dependent cellular cytotoxicity and immunogenic cell death. This provides a rationale for combination therapy utilizing trastuzumab and ICI [132]. Currently, KEYNOTE-811 (NCT03615326) is ongoing to evaluate the role of pembrolizumab in combination with standard 1L HER2-targeted chemotherapy with trastuzumab in HER2-positive advanced GC compared to placebo.

Increasing evidence has shown that angiogenic factors change the tumor microenvironment to an immunosuppressive state. The relevant mechanisms include suppression of dendritic cell maturation and recruitment of regulatory T-cells, myeloid-derived suppressor cells, and tumor-associated macrophages [133]. Antiangiogenic agents have an immune-modulating effect that restores antitumor activity by alteration of the tumor microenvironment [134]. Recent early-phase clinical trials have shown promising efficacy in combining PD-1/PD-L1 inhibitors and VEGFR2-targeted drugs for advanced GC. In the phase Ib KEYNOTE-098 trial (NCT02443324), the safety and promising efficacy of 1L pembrolizumab plus ramucirumab was demonstrated [135]. In another phase I/II trial (NivoRam: NCT02999295), 2L nivolumab plus ramucirumab showed mild toxicity and promising antitumor activity [136]. A second-line three-drug combination consisting of paclitaxel, ramucirumab, and nivolumab has been tested in a Japanese phase I/II trial (UMIN000025947) and demonstrated manageable safety and promising efficacy [137]. Currently, other phase II trials are testing similar combination therapies. The RAP trial in Germany (NCT03966118) is checking the combination of paclitaxel, ramucirumab, and avelumab, while the SEQUEL trial in the USA (NCT04069273) is evaluating that of paclitaxel, ramucirumab, and pembrolizumab.

Regorafenib, a multi-targeted TKI, is said to enhance antitumor immunity via macrophage modulation [138]. In a recently reported Japanese phase Ib trial, (EPOC1603: NCT03406871), combining nivolumab with regorafenib demonstrated encouraging antitumor effects in advanced GC, which was refractory to standard chemotherapies (median PFS 5.8 months, ORR 44%) [139]. Another multi-targeted TKI, lenvatinib, was also evaluated in combination with pembrolizumab in a Japanese phase II trial (EPOC1706: NCT03609359), which showed a promising efficacy (ORR 69%) [140].

## 7. Concluding Remarks

Here, we extensively summarize and update information on global developments in the treatment of GC, focusing on systemic chemotherapy. In addition, we pointed out several geographical differences in epidemiology, biological profiles, and standard of care between the East and the West (Table 5).

While favorable outcomes of surgery improve curability, especially in Asian countries, adjuvant treatment is necessary to cure resectable disease. In each region, there has been robust effort to develop more effective adjuvant treatments, which have been established as regional standards. Sequential chemotherapy is a standard non-curative strategy for advanced stage GC. Besides powerful combination regimens, several targeted drugs have recently been evaluated. However, the highly heterogeneous biology of GC has hampered the success of most of these clinical trials, except for those testing trastuzumab and ramucirumab. To overcome this issue, a combination strategy simultaneously inhibiting several oncogenic pathways and/or enriched patient selection is needed. One of the greatest recent clinical successes has been the introduction of ICIs (PD-1 inhibitors). Beyond the approved use of ICIs as a monotherapy in heavily treated advanced GC, recent development has been focused on the 1L use of ICIs and combination use with targeted drugs and other ICIs. These active developments are expected to lead the realization of needed advances to effectively treat GC.

## Figures and Tables

**Figure 1 cancers-12-01100-f001:**
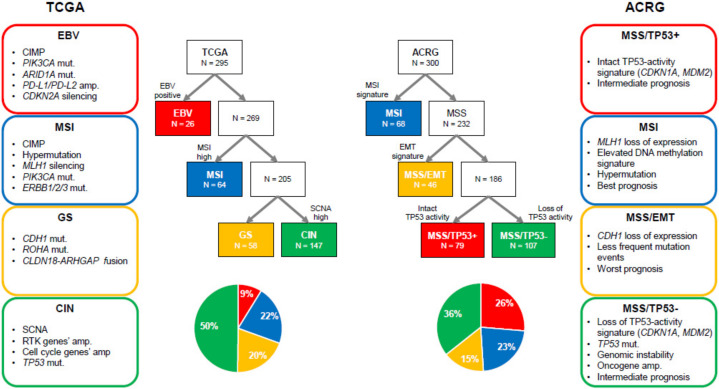
Two different molecular classifications of gastric cancer. ACRG, Asian Cancer Research Group; amp, amplification; CIMP, CpG island methylation phenotype; CIN, chromosomal instability; EBV, Epstein-Barr virus; EMT, epithelial-mesenchymal transition; GS, genomically stable; MSI, microsatellite instability; MSS, microsatellite stable; mut, mutation; RTK, receptor tyrosine kinase; SCNA, somatic copy number aberrations; TCGA, The Cancer Genome Atlas.

**Table 1 cancers-12-01100-t001:** Pivotal phase III (or II/III) trials of adjuvant therapy in gastric cancer.

Study	Year	Region	Phase	Setting	N	Subject	Lymphadenectomy	Treatment Arm	PE	Result
INT0116	2001	US	III	Post	603	GC after curative resection	D0: 54% D1: 36% D2: 10%	CRT (5-FU/FA) Surgery alone	RFS/OS	Positive
CALGB80101	2017	US	III	Post	546	GC after curative resection	NA	CRT (ECF) CRT (5-FU/FA)	OS	Negative
MAGIC	2006	UK	III	Peri	503	Resectable GC (including the lower esophagus)	D1: 18% D2: 38%	ECF Surgery alone	OS	Positive
MRC ST03	2017	UK	II/III	Peri	1063	Resectable GC (including the lower esophagus)	Not available	ECF + bevacizumab ECF	OS	Negative
FLOT4	2019	Germany	II/III	Peri	716	Resectable GC	D1: 2% D2: 55%	FLOT ECF (ECX)	OS	Positive
CRITICS	2018	Netherlands	III	Peri	788	Resectable GC	D1+: 79% D2: 6%	Pre ECX (EOX) + Post CRT (XP) Peri ECX (EOX)	OS	Negative
ACTS-GC	2007	Japan	III	Post	1059	GC after curative resection	D2: 94% D3: 6%	S-1 Surgery alone	OS	Positive
START2	2019	Japan	III	Post	915	GC after curative resection (Only stage III cases)	D2: 100%	S-1+DTX S-1	RFS	Positive
CLASSIC	2012	Korea	III	Post	1035	GC after curative resection	D2: 100%	CAPOX Surgery alone	DFS	Positive
ARTIST	2012	Korea	III	Post	458	GC after curative resection	D2: 100%	XP XP + CRT (X)	DFS	Negative
ARTIST-II	2019	Korea	III	Post	538	GC after curative resection (Only LN-positive cases)	D2: 100%	SOX + CRT (S-1) SOX S-1	DFS	Positive Negative
PRODIGY	2019	Korea	III	Peri	530	Resectable GC (cT2-3/N+, or T4)	D2: 98%	Pre DOS + Post S-1 Post S-1	PFS	Positive
RESOLVE	2019	China	III	Peri/Post	1022	Resectable GC (cT4a/N+, or cT4b)	D2: 100%	Peri SOX Post SOX (non-inferiority) Post CAPOX	DFS	Positive

CAPOX, capecitabine + oxaliplatin; CRT, chemoradiotherapy; DFS, disease-free survival; DOS, docetaxel + oxaliplatin + S-1; DTX, docetaxel; ECF, epirubicin + cisplatin + 5-FU; ECX, epirubicin + cisplatin + capecitabine; EOX, epirubicin + oxaliplatin + capecitabine; FA, folinic acid; FLOT, 5-FU + leucovorin + oxaliplatin + docetaxel; GC, gastric cancer; LN, lymph node; OS, overall survival; PE, primary endpoint; Peri, peri-operative; Post, post-operative; Pre, pre-operative; RFS, relapse-free survival; SOX, S-1 + oxaliplatin; X, capecitabine; XP, capecitabine + cisplatin.

**Table 2 cancers-12-01100-t002:** Ongoing randomized phase II and III trials of adjuvant therapy in gastric cancer.

Study	Region	Phase	Setting	N	Subject	Treatment Arm	PE
INNOVATION (NCT02205047)	Europe	rII	Peri	220	HER2-positive resectable GC	XP (FP) + trastuzumab + pertuzumab XP (FP) + trastuzumab XP (FP)	Near pCR rate
TOPGEAR (NCT01924819)	Australia Canada Europe	II/III	Peri	620	Resectable GC	Peri ECF (ECX, EOX) or FLOT + Pre CRT (5-FU or X) Peri ECF (ECX, EOX) or FLOT	OS
PETRARCA (FLOT6) (NCT02581462)	Germany	II/III	Peri	81	HER2-positive resectable GC	FLOT + trastuzumab + pertuzumab FLOT	DFS
RAMSES (FLOT7) (NCT02661971)	Germany	II/III	Peri	908	HER2-negative resectable GC	FLOT + ramucirumab FLOT	OS
DANTE (NCT03421288)	Germany	rII	Peri	295	Resectable GC	FLOT + atezolizumab FLOT	DFS PFS
IMAGINE (NCT04062656)	Germany	rII	Peri	88	Resectable GC	Nivolumab Nivolumab + ipilimumab Nivolumab + relatlimab (anti-LAG3) FLOT	pCR rate
CRITICS-II (NCT02931890)	Netherlands	rII	Pre	207	Resectable GC	DOC DOC + CRT (CBDCA+PTX) CRT (CBDCA+PTX)	EFS
ATTRACTION-05 (NCT03006705)	Asia	III	Post	700	GC after curative resection (pStage III)	S-1 or CAPOX + nivolumab S-1 or CAPOX + placebo	RFS
NAGISA (JCOG1509) (UMIN000024065)	Japan	III	Peri	470	Resectable GC (cT3-4/N1-3)	Pre SOX + Post S-1 or DS Post S-1 or DS	OS
Trigger (JCOG1301C) (UMIN000016920)	Japan	rII	Peri	130	HER2-positive resectable GC with extensive LN metastasis	Pre SP + trastuzumab + Post S-1 Pre SP + Post S-1	OS
Neo-CRAG (NCT01815853)	China	III	Peri	620	Resectable GC (cT3/N2-3, cT4a/N+, or cT4b)	Pre CRT (CAPOX) + Post CAPOX Peri CAPOX	DFS
PREACT (NCT03013010)	China	III	Peri	682	Resectable GC (cStage IIB and III)	Peri SOX + Pre CRT (S-1) Peri SOX	DFS
RESCUE-GC (NCT02867839)	China	III	Post	564	GC after curative resection (pStage II and IIIA)	SOX S-1	DFS
KEYNOTE-585 (NCT03221426)	International	III	Peri	860	Resectable GC	XP (FP) + pembrolizumab XP (FP) + placebo FLOT + pembrolizumab FLOT + placebo	OS EFS pCR rate

CAPOX, capecitabine + oxaliplatin; CBDCA, carboplatin; CRT, chemoradiotherapy; DFS, disease-free survival; DOC, docetaxel + oxaliplatin + capecitabine; DS, docetaxel + S-1; ECF, epirubicin + cisplatin + 5-FU; ECX, epirubicin + cisplatin + capecitabine; EFS, event-free survival; EOX, epirubicin + oxaliplatin + capecitabine; FLOT, 5-FU + leucovorin + oxaliplatin + docetaxel; FP, 5-FU + cisplatin; GC, gastric cancer; LN, lymph node; rII, randomized phase II; OS, overall survival; pCR, pathological complete remission; Peri, peri-operative; PFS, progression-free survival; Post, post-operative; Pre, pre-operative; PTX, paclitaxel; RFS, relapse-free survival; SOX, S-1 + oxaliplatin; SP, S-1 + cisplatin; X, capecitabine; XP, capecitabine + cisplatin.

**Table 3 cancers-12-01100-t003:** Phase III (or II/III) trials of targeted therapy in advanced gastric cancer.

Study	Year	Region	Phase	Target	Drug	Line	Subject	N	TREATMENT ARM	PE	Result
ToGA	2010	International	III	HER2	Trastuzumab	1L	HER2 positive (IHC3+, FISH+)	594	XP (FP) + trastuzumab XP (FP)	OS	Positive
TRIO-013/LOGiC	2016	International	III	HER1/2	Lapatinib	1L	HER2 positive (FISH+)	545	CAPOX + lapatinib CAPOX + placebo	OS	Negative
TyTAN	2014	Asia	III	HER1/2	Lapatinib	2L	HER2 positive (FISH+)	261	PTX + lapatinib PTX	OS	Negative
JACOB	2018	International	III	HER2	Pertuzumab	1L	HER2 positive (IHC3+, IHC2+/FISH+)	780	XP (FP) + trastuzumab + pertuzumab XP (FP) + trastuzumab + placebo	OS	Negative
GATSBY	2017	International	II/III	HER2	T-DM1	2L	HER2 positive (IHC3+, IHC2+/FISH+)	345	T-DM1 Taxane (DTX or PTX)	OS	Negative
ASLAN001-012 (NCT03130790)	Ongoing	Mainly Asia	II/III	HER1/2/4	Varlitinib	1L	HER1/2 co-expressing	400	mFOLFOX6 + varlitinib mFOLFOX6 + placebo	OS	Not yet
EXPAND	2013	International	III	HER1	Cetuximab	1L	All	904	XP + cetuximab XP	PFS	Negative
REAL3	2013	UK	III	HER1	Panitumumab	1L	All	553	mEOC + panitumumab EOC	OS	Negative
RILOMET-1	2017	International without Asia	III	HGF	Rilotumumab	1L	MET positive HER2 negative	609	ECX + rilotumumab ECX + placebo	OS	Negative
METGastric	2017	International	III	MET	Onartuzumab	1L	MET positive HER2 negative	562	mFOLFOX6 + onartuzumab mFOLFOX6 + placebo	OS	Negative
FIGHT (NCT03694522)	Ongoing	International	III	FGFR2b	Bemarituzumab (FPA144)	1L	FGFR2b overexpression or *FGFR2* amplification HER2 negative	548	mFOLFOX6+bemarituzumab mFOLFOX6 + placebo	OS	Not yet
GRANITE-1	2013	International	III	mTOR	Everolimus	2L/3L	All	656	Everolimus + BSC Placebo + BSC	OS	Negative
RADPAC	2017	Germany	III	mTOR	Everolimus	2L/3L /4L	All	300	PTX + everolimus PTX + placebo	OS	Negative
GOLD	2017	Asia	III	PARP	Olaparib	2L	All	525	PTX + olaparib PTX + placebo	OS	Negative
PARALLEL 303 (NCT03427814)	Ongoing	International	III	PARP	Pamiparib (BGB-290)	1L	All	540	Pamiparib (maintenance) Placebo (maintenance)	PFS	Not yet
BRIGHTER	2018	International	III	STAT3	Napabucasin	2L	All	714	PTX + napabucasin PTX + placebo	OS	Negative
GAMMA-1	2019	International without Asia	III	MMP9	Andecaliximab	1L	HER2 negative	432	mFOLFOX6 + andecaliximab mFOLFOX6 + placebo	OS	Negative
SPOTLIGHT (NCT03504397)	Ongoing	International	III	Claudin 18.2	Zolbetuximab (IMAB362)	1L	Claudin 18.2 positive HER2 negative	550	mFOLFOX6 + zolbetuximab mFOLFOX + placebo	PFS	Not yet
GLOW (NCT03653507)	Ongoing	International	III	Claudin 18.2	Zolbetuximab (IMAB362)	1L	Claudin 18.2 positive HER2 negative	500	CAPOX + zolbetuximab CAPOX + placebo	PFS	Not yet
AVAGAST	2011	International	III	VEGFA	Bevacizumab	1L	All	774	XP (FP) + bevacizumab XP (FP) + placebo	OS	Negative
AVATAR	2015	China	III	VEGFA	Bevacizumab	1L	All	202	XP + bevacizumab XP + placebo	OS	Negative
RAINBOW	2014	International	III	VEGFR2	Ramucirumab	2L	All	665	PTX + ramucirumab PTX + placebo	OS	Positive
REGARD	2014	International	III	VEGFR2	Ramucirumab	2L	All	355	Ramucirumab + BSC Placebo + BSC	OS	Positive
RAINFALL	2019	International	III	VEGFR2	Ramucirumab	1L	HER2 negative	645	XP (FP) + ramucirumab XP (FP) + placebo	PFS	Positive
RINDBeRG (UMIN000023065)	Ongoing	Japan	III	VEGFR2	Ramucirumab	3L	All	400	IRI + ramucirumab (beyond progression) IRI	OS	Not yet
ARMANI (NCT02934464)	Ongoing	Italy	III	VEGFR2	Ramucirumab	1L	HER2 negative	280	PTX + ramucirumab (switch maintenance) FOLFOX4, mFOLFOX6. or CAPOX	PFS	Not yet
HENGRUI 20101208	2016	China	III	VEGFR2	Apatinib	≥ 3L	All	267	Apatinib Placebo	OS PFS	Positive
ANGEL	2019	International	III	VEGFR2	Apatinib	≥ 3L	All	460	Apatinib + BSC Placebo + BSC	OS	Negative
TJCC006 (NCT03598348)	Ongoing	China	III	VEGFR2	Apatinib	1L	HER2 negative	288	Apatinib + X (maintenance after CAPOX) Apatinib (maintenance after CAPOX) Observation (after CAPOX)	PFS	Not yet
FRUTIGA (NCT03223376)	Ongoing	China	III	VEGFR1/2/3	Fruquintinib	2L	All	544	PTX + fruquintinib PTX + placebo	OS	Not yet
INTEGRATE II (NCT02773524)	Ongoing	International	III	Multi-target	Regorafenib	≥ 3L	All	350	Regorafenib Placebo	OS	Not yet

BSC, best supportive care; CAPOX, capecitabine + oxaliplatin; ECX, epirubicin + cisplatin + capecitabine; FP, 5-FU + cisplatin; IRI, irinotecan; mEOC, modified-dose EOC (epirubicin + oxaliplatin + capecitabine); mFOLFOX6, modified FOLFOX6 (5-FU + leucovorin + oxaliplatin); OS, overall survival; PFS, progression-free survival; PTX, paclitaxel; XP, capecitabine + cisplatin.

**Table 4 cancers-12-01100-t004:** Ongoing clinical trials of immune checkpoint inhibitors in advanced gastric cancer.

Study	Region	Phase	Drugs (Target)	Line	N	Subject	Treatment Arm	PE
AK104 (NCT03852251)	China	I/II	AK104 (PD-1/CTLA-4)	1L	112	HER2 negative	AK104 CAPOX+AK104	ORR
CP-MGAH22-05 (NCT02689284)	International	I/II	Pembrolizumab (PD-1) Margetuximab (HER2)	≥ 2L	95	HER2 positive	Margetuximab + pembrolizumab	ORR DOR
KEYNOTE-659 (NCT03382600)	Japan	II	Pembrolizumab (PD-1)	1L	90	PD-L1 positive HER2 negative	SOX + pembrolizumab (Cohort 1) SP + pembrolizumab (Cohort 2)	ORR
EPOC1706 (NCT03609359)	Japan	II	Pembrolizumab (PD-1) Lenvatinib (multi-target)	-	29	All	Lenvatinib + pembrolizumab	ORR
ESR-15-11655 (NCT03579784)	Korea	II	Durvalumab (PD-1) Olaparib (PARP)	2L	40	All	PTX + olaparib + durvalumab	DCR
NCC2070 (NCT04140318)	China	II	Sintilimab (PD-1)	2L	60	All	Nab-PTX + sintilimab	ORR
ASGARD (NCT04089657)	China	II	Sintilimab (PD-1)Apatinib (VEGFR2)	≥ 3L	40	All	Apatinib + sintilimab	DCR
RiME (NCT03995017)	US	II	Nivolumab (PD-1) Rucaparib (PARP) Ramucirumab (VEGFR2)	2L/3L	61	All	Rucaparib + ramucirumab + nivolumab Rucaparib + ramucirumab	ORR
16-937 (NCT02954536)	US	II	Pembrolizumab (PD-1) Trastuzumab (HER2)	1L	37	HER2 positive	XP (CAPOX) + trastuzumab + pembrolizumab	PFS
RAP (AIO-STO-0218) (NCT03966118)	Germany	II	Avelumab (PD-1) Ramucirumab (VEGFR2)	2L	59	All	PTX + ramucirumab + avelumab	OS
INTEGA (AIO-STO-0217) (NCT03409848)	Germany	rII	Nivolumab (PD-1) Ipilimumab (CTLA-4) Trastuzumab (HER2)	1L	97	HER2 positive	Nivolumab + ipilimumab + trastuzumab mFOLFOX6 + trastuzumab + nivolumab	OS
MOONLIGHT (AIO-STO-0417) (NCT03647969)	Germany	rII	Nivolumab (PD-1) Ipilimumab (CTLA-4)	1L	118	HER2 negative	mFOLFOX6 + nivolumab + ipilimumab mFOLFOX6	PFS
DURIGAST (PRODIGE59-FFCD1707) (NCT03959293)	France	rII	Durvalumab (PD-L1) Tremelimumab (CTLA-4)	2L	105	All	FOLFIRI + durvalumab + tremelimumab FOLFIRI + durvalumab	PFS
SEQUEL (NCT04069273)	US	rII	Pembrolizumab (PD-1) Ramucirumab (VEGFR2)	≥ 2L	58	All	PTX + ramucirumab + pembrolizumab (with patient-tailored algorithm) PTX + ramucirumab + pembrolizumab	ORR
CA224-060 (NCT03662659)	International without Asia	rII	Nivolumab (LAG-3) Relatlimab (PD-1)	1L	250	HER2 negative	CAPOX (FOLFOX, SOX) + nivolumab + relatlimab CAPOX (FOLFOX, SOX) + nivolumab	ORR
FRACTION-GC (NCT02935634)	International without Asia	rII	Nivolumab (PD-1) Ipilimumab (CTLA-4) Relatlimab (LAG-3) BMS-986205 (IDO1) Rucaparib (PARP)	-	600	All	Nivolumab + relatlimab Nivolumab + BMS-986205 Nivolumab + rucaparib Ipilimumab + rucaparib Nivolumab + ipilimumab + rucaparib Nivolumab + ipilimumab	ORR DOR PFS
ATTRACTION-04 (NCT02746796)	Asia	II/III	Nivolumab (PD-1)	1L	680	HER2 negative	CAPOX (SOX ) + nivolumab CAPOX (SOX) + placebo	OS PFS
MAHOGANY (NCT04082364)	US	II/III	MGA012 (PD-1) MGD013 (PD-1/LAG-3) Margetuximab (HER2)	1L	850	Cohort A: HER2/PD-L1 positive Cohort B: HER2 positive	Margetuximab + MGA012 CAPOX (mFOLFOX6) + margetuximab + MGA012 CAPOX (mFOLFOX6) + margetuximab + MGD013 CAPOX (mFOLFOX6) + margetuximab CAPOX (mFOLFOX6) + trastuzumab	Cohort A: ORR Cohort B: OS
KEYNOTE-063 (NCT03019588)	Asia	III	Pembrolizumab (PD-1)	2L	360	PD-L1 positive	Pembrolizumab PTX	OS PFS
GEMSTONE-303 (NCT03802591)	China	III	CS1001 (PD-L1)	1L	480	HER2 negative	CAPOX + CS1001 CAPOX + placebo	OS PFS
SHR-1210-III-311 (NCT03813784)	China	III	SHR-1210 (PD-1) Apatinib (VEGFR2)	1L	568	HER2 negative	CAPOX + SHR-1210 followed by apatinib + SHR-1210 CAPOX	OS
CIBI308E301 (NCT03745170)	China	III	Sintilimab (PD-1)	1L	650	HER2 negative	CAPOX + sintilimab CAPOX + placebo	OS
CheckMate 649 (NCT02872116)	International	III	Pembrolizumab (PD-1) Ipilimumab (CTLA-4)	1L	2005	HER2 negative	Nivolumab + ipilimumab CAPOX (FOLFOX) + nivolumab CAPOX (FOLFOX)	OS PFS
KEYNOTE-811 (NCT03615326)	International	III	Pembrolizumab (PD-1) Trastuzumab (HER2)	1L	732	HER2 positive	FP (CAPOX, SOX) + trastuzumab + pembrolizumab FP (CAPOX, SOX) + trastuzumab + placebo	OS PFS
KEYNOTE-859 (NCT03675737)	International	III	Pembrolizumab (PD-1)	1L	780	HER2 negative	FP (CAPOX) + pembrolizumab FP (CAPOX)+placebo	OS PFS
BGB-A317-305 (NCT03777657)	International	III	Tislelizumab (PD-1)	1L	720	HER2 negative	CAPOX (FP) + tislelizumab CAPOX (FP)+placebo	OS PFS

CAPOX, capecitabine + oxaliplatin; DCR, disease control rate; DOR, duration of response; FOLFIRI, 5-FU + leucovorin + irinotecan; FP, 5-FU + cisplatin; mFOLFOX6, modified FOLFOX6 (5-FU + leucovorin + oxaliplatin); ORR, objective response rate; OS, overall survival; PFS, progression-free survival; PTX, paclitaxel; SOX, S-1 + oxaliplatin; SP, S-1 + cisplatin; XP, capecitabine + cisplatin.

**Table 5 cancers-12-01100-t005:** Differences in epidemiology, biology, and clinical practice between the East and the West.

Variation	East	West
Incidence (percentage in the world)	Asia: 48.4%	Europe: 23.4% North America: 13.2%
Localization of the primary legion	Distal third of the stomach	Proximal third of the stomach
Stage at the diagnosis	Early stage	Late stage
Overall survival	Better	Worse
Nationwide screening program	Present (Japan, South Korea, Matsu Islands in Taiwan)	Absent
Endoscopic resection for early stage GC	Common	Uncommon
Prevalence of *Helicobacter pylori* infection	Asia: 54.7%	Europe: 47.0% North America: 37.1%
Immune profiles	Enriched neutrophil marker (CD66b) Enriched pan-leukocyte maker (CD45) Enriched immunosuppressive T-regulatory cell marker (FOXP3)	Enriched T-cell signatures (CD28, CTLA-4 signaling) Enriched T-cell markers (CD3, CD45R0, CD8)
Surgery for resectable GC	D2 gastrectomy	D1 gastrectomy
Adjuvant treatment	Asia: post-operative chemotherapy (CAPOX for 6M, S-1 for 1Y, S-1+DTX for Stage III in Japan)	Europe: peri-operative chemotherapy (FLOT) US: post-operative chemoradiotherapy
1L treatment for advanced GC	HER2-negative GC: platinum + fluoropyrimidine HER2-positive GC: platinum + fluoropyrimidine + trastuzumab	
2L treatment for advanced GC	Paclitaxel + ramucirumab Single agent: paclitaxel, docetaxel, irinotecan, ramucirumab	
3L treatment for advanced GC	Nivolumab Irinotecan Apatinib (only in China) FTD/TPI	FTD/TPI Pembrolizumab for PD-L1-positive (CPS ≥ 1%) GC
Post-progression treatment	1L to 2L: 69–85% (in Japan) 2L to 3L: 58–90% (in Japan)	1L to 2L: 11–59% 2L to 3L: 8–29%

CAPOX, capecitabine + oxaliplatin; CPS, combined positive score; DTX, docetaxel; FLOT, 5-FU + leucovorin + oxaliplatin + docetaxel; FTD/TPI, trifluridine/tipiracil; GC, gastric cancer; 1L, first-line; 2L, second-line; 3L, third-line.

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
