# Peer review of "Recent Developments of Systemic Chemotherapy for Gastric Cancer"

_cancers, 2020, doi:10.3390/cancers12051100_

Round 1

Reviewer 1 Report

It was a pleasure to review the work of Dr. Hiroyuki Arai and Dr. Takako Eguchi Nakajima.

The treatment landscape of gastric cancer is rapidly changing and the authors have done a commendable job of reviewing the latest data. The work is extensive and does count as a valuable literature for review. I recommend publication of their work subject to the necessary correction as detailed below:

Citation #1: please provide the citation from a primary source.

Despite recent advances in multimodal treatment, the prognosis remains poor for both localized and advanced GC - With the FLOT data, prognosis for localized GC can no longer be called poor. Please correct.

Recently, many studies have focused on unveiling the complicated molecular background of GC, developing more effective treatment strategies. .  Please rephrase as there has  no effective treatment has been developed based on molecular characterization yet. It does hold a promise and it would be appropriate to say  development of more effective treatment strategy based on new molecular data is possible in future.

Table 1: CIN tumor are molecularly similar to MSS/PT53+ and not MSS/TP53- ( loss of p53 is common with both).

6.2.3. Other targeted therapies section: Please include the zolbetuximab , promising phase 2 data and phase 3 studies ongoing.

Author Response

Reviewer’s Comments:

Reviewer 1:

The treatment landscape of gastric cancer is rapidly changing and the authors have done a commendable job of reviewing the latest data. The work is extensive and does count as a valuable literature for review. I recommend publication of their work subject to the necessary correction as detailed below:

Thank you for your kind comments. We made some modifications according to the points your indicated.

Major Points:

1. Citation #1: please provide the citation from a primary source.

We changed and provided the Citation #1 as primary source.

2. Despite recent advances in multimodal treatment, the prognosis remains poor for both localized and advanced GC - With the FLOT data, prognosis for localized GC can no longer be called poor. Please correct.

     We changed the sentence as below.

Before

After

Page 1

Line 38

(Section 1)

the prognosis remains poor for both localized and advanced GC

the prognosis remains poor for advanced GC

3. Recently, many studies have focused on unveiling the complicated molecular background of GC, developing more effective treatment strategies. Please rephrase as there has no effective treatment has been developed based on molecular characterization yet. It does hold a promise and it would be appropriate to say development of more effective treatment strategy based on new molecular data is possible.

 We changed the sentence as below.

Before

After

Page 2

Line 44-46

(Section 1)

Recently, many studies have focused on unveiling the complicated molecular background of GC, developing more effective treatment strategies.

Although no effective treatments have been developed based on molecular characterization yet, development of more effective treatment strategy based on new molecular data could be possible in future.

4. Table 1: CIN tumor are molecularly similar to MSS/PT53+ and not MSS/TP53- (loss of p53 is common with both).

  We also received comments from reviewer 2 about exposing molecular classifications. We thought the way to explanation of this subject should be easier to understand. Therefore, we replaced the Table 1 with new Figure 1.

5. 6.2.3. Other targeted therapies section: Please include the zolbetuximab, promising phase 2 data and phase 3 studies ongoing.

  Thank you for your pointing out. We included the data from FAST trial which showed the promising efficacy of zolbetuximab in combination with first-line EOX. Also, we introduced the ongoing phase III trials of this drug (SPOTLIGHT and GLOW trials)

Before

After

Page 15

Line 431-436
(Section 6.2.3.)

None

Recently, promising results were reported in a randomized phase II trial (FAST) showing tight junction protein claudin 18.2 (CLDN18.2)-targeted zolbetuximab improved PFS (HR 0.47; 95 % CI 0.31–0.70; p = 0.0001) and OS (HR 0.51; 95 % CI 0.36–0.73; p = 0.0001) in combination with 1L EOX in patients with CLDN18.2-positive GC [114]. Currently, two phase III trials are ongoing to test the 1L use of zolbetuximab in combination with mFOLFOX6 (SPOTLIGHT: NCT03504397) or CAPOX (GLOW: NCT03653507).

Reviewer 2 Report

The aim of the manuscript is to describe the different strategy evaluated for gastric cancer. The paper is well written and documented on many of the phase 3 trials reported.

Major remarks:

The paper focus on chemotherapy using intravenous treatment. But, many new treatment options exist and have been identified to be able to change the course of the disease.

As by example HIPEC in case of complete cytoreductive surgery – J Clin Oncol. 2019 Aug 10;37(23):2028-2040. doi: 10.1200/JCO.18.01688. Epub 2019 May 14. (Cytoreductive Surgery With or Without Hyperthermic Intraperitoneal Chemotherapy for Gastric Cancer With Peritoneal Metastases (CYTO-CHIP study): A Propensity Score Analysis. by Bonnot PE, et al).

Or PIPAC for a new drug delivery solution (Lancet Oncol. 2019 Jul;20(7):e368-e377. doi: 10.1016/S1470-2045(19)30318-3. Pressurised intraperitoneal aerosol chemotherapy: rationale, evidence, and potential indications. Alyami M, et al)

Or Radiomics that can offer new solution for driving surgery or for identification of different biological information (Eur Radiol. 2020 Apr;30(4):2324-2333. doi: 10.1007/s00330-019-06621-x. Epub 2020 Jan 17. Dual-energy CT-based deep learning radiomics can improve lymph node metastasis risk prediction for gastric cancer. Li J, et al.)

For all these reasons, the tittle of the paper must be change to explain that the paper is limited to intravenous drugs delivery and not more.

The paper is not didactic.

Some major figures are lacking – that can expose the different subtypes of gastric cancer and some than can expose the different drug regiments with survival curves.

It is a real hard work, but the paper actually is just a catalogue and a real work of synthesis should be propose.

The two authors are experts but exposing and explaining is not so easy. A flow chart could be proposed, maybe regarding the different tumor genetic's information and places as western or Asia areas.

Author Response

Reviewer’s Comments:

Reviewer 2:

The aim of the manuscript is to describe the different strategy evaluated for gastric cancer. The paper is well written and documented on many of the phase 3 trials reported.

Major Points:

1. The paper focus on chemotherapy using intravenous treatment. But, many new treatment options exist and have been identified to be able to change the course of the disease.

As by example HIPEC in case of complete cytoreductive surgery – J Clin Oncol. 2019 Aug 10;37(23):2028-2040. doi: 10.1200/JCO.18.01688. Epub 2019 May 14. (Cytoreductive Surgery With or Without Hyperthermic Intraperitoneal Chemotherapy for Gastric Cancer With Peritoneal Metastases (CYTO-CHIP study): A Propensity Score Analysis. by Bonnot PE, et al).

Or PIPAC for a new drug delivery solution (Lancet Oncol. 2019 Jul;20(7):e368-e377. doi: 10.1016/S1470-2045(19)30318-3. Pressurised intraperitoneal aerosol chemotherapy: rationale, evidence, and potential indications. Alyami M, et al)

Or Radiomics that can offer new solution for driving surgery or for identification of different biological information (Eur Radiol. 2020 Apr;30(4):2324-2333. doi: 10.1007/s00330-019-06621-x. Epub 2020 Jan 17. Dual-energy CT-based deep learning radiomics can improve lymph node metastasis risk prediction for gastric cancer. Li J, et al.)

For all these reasons, the tittle of the paper must be change to explain that the paper is limited to intravenous drugs delivery and not more.

  Thank you for your important comment. As you mentioned, there are many treatment options besides intravenously delivered chemotherapies. Therefore, we changed the title of this review article to ‘’Recent developments of systemic chemotherapy for gastric cancer’’.

2. The paper is not didactic.

Some major figures are lacking – that can expose the different subtypes of gastric cancer and some than can expose the different drug regiments with survival curves.

It is a real hard work, but the paper actually is just a catalogue and a real work of synthesis should be proposed.

We appreciate your critical comments. With all due respect, we can understand your pointing out and can learn about how we should write a didactic review from your comments. We actually focused on broadly to introduce novel information based on the recently reported phase III trials, which might result in impressing as just a catalogue. However there have been many advances in this area including novel combination, targeted-drug, and immunotherapies to update, accounting for most part of our article.

Instead, we shored up explanation with additional figure (Figure 1) to expose two different molecular classifications established by TCGA and ACRG. We believe this is helpful for easier understanding of the subject. As for the figure with survival curves, we couldn’t resynthesize any published clinical trials survival data on Kaplan-Meier curve.

3. The two authors are experts but exposing and explaining is not so easy. A flow chart could be proposed, maybe regarding the different tumor genetics information and places as western or Asia areas.

 ã€€Thank you for your important proposal. We made up an additional table (Table 5) to expose the geographical differences in epidemiology, biology, and clinical practice between the East and the West. This table is for a summary of geographical differences that we explained in several sections of this article. To make up this table, we shored up some explanations on regional differences as below, all of which are not included in the first draft. Also, we modified some sentences included the part of concluding remarks as below.

Before

After

Page 2

Line 75-80

(Section 2)

None

Tumors located in the proximal third of the stomach are more common in Western countries [18,19]. Proximal tumors are associated with more advanced stage at presentation, a larger tumor size, and poorly differentiated histology [20]. This may account for the worse survival in the West.

H. pylori infection increases cancer risk especially for intestinal-type distal carcinoma [21]. Prevalence of H. pylori in Asia is 54.7%, which is higher than in Europe (47.0%) or in North America (37.1%) [22].

Page 12

Line 287-289

(Section 6)

None

Of note, the rate of receiving post-progression therapy is much higher in Japan than in the Western countries (2L: 69 %–85 % vs. 11 %–59 %; 3L: 58 %–90 % vs. 8 %–29 %) [68-72].

Page 26

Line 570-573

(Section 7)

While GC is a significant global health problem, its epidemiology, biological profiles, and standard of care are geographically distinct.

Here, we extensively summarize and update information of global developments in the treatment of GC, focusing on systemic chemotherapy. In addition, we pointed out several geographical differences in epidemiology, biological profiles, and standard of care between the East and the West (Table 5).